# Lupus Erythematosus Quality of Life Questionnaire (LEQoL): Development and Psychometric Properties

**DOI:** 10.3390/ijerph17228642

**Published:** 2020-11-20

**Authors:** Elena Castellano-Rioja, María del Carmen Giménez-Espert, Ana Soto-Rubio

**Affiliations:** 1Department of Nursing, Faculty of Nursing, Catholic University of Valencia, 46001 Valencia, Spain; elena.castellano@ucv.es; 2Department of Nursing, Faculty of Nursing and Chiropody, University of Valencia, 46010 Valencia, Spain; 3Personality, Assessment and Psychological Treatments Department, Faculty of Psychology, University of Valencia, 46010 Valencia, Spain; ana.soto@uv.es

**Keywords:** instrument, lupus erythematosus, cutaneous, lupus erythematosus, systemic, quality of life

## Abstract

Lupus erythematosus (LE) affects patients’ quality of life. Nevertheless, no instrument has been developed to assess the quality of life in systemic lupus erythematosus (SLE) and cutaneous lupus erythematosus (CLE) patients. This study aims to develop and psychometrically test the “Quality of Life of Patients with Lupus Erythematosus Instrument” (LEQoL) and study the quality of life of these patients. Finally, percentiles for interpreting scores of LEQoL in patients with LE are provided. This study is cross-sectional, with a sample of 158 patients recruited from a lupus association for the psychometric evaluation of the final version of LEQoL. The scale’s reliability was assessed by Cronbach’s alpha, composite reliability (CR), and average variance extracted (AVE). Validity was examined through exploratory factorial analyses (EFA) and confirmatory factorial analyses (CFA). The definitive model, composed of 21 items grouped into five factors, presented good psychometric properties. Mean levels of quality of life were observed in patients with systemic LE, with higher values in patients with cutaneous LE. The LEQoL instrument is a useful tool for assessing the quality of life of patients with LE, allowing the evaluation of current clinical practices, the identification of educational needs, and the assessment of the effectiveness of interventions intended to improve the quality of life of patients with LE, SLE, and CLE.

## 1. Introduction

Lupus erythematosus (LE) is an autoimmune, inflammatory, multisystemic, and chronic disease of unknown cause [1]. LE was classified as a rare disease until a few years ago and is now considered an infrequent disease, mainly due to increased prevalence, improved diagnostic criteria, and the development of screening tests [2]. Although there are indeed other classifications of the disease, the most common would be that which considers systemic lupus erythematosus (SLE) and cutaneous lupus erythematosus (CLE) [1]. For the relationship between the two types of lupus, 80% of patients with SLE develop skin manifestations, and approximately 25% have skin lesions in their first manifestation [3].

The Lupus Foundation of America estimates that 1.5 million Americans and at least five million people worldwide have a form of lupus [4]. Concerning SLE, in general, European countries have a lower prevalence, whereas Asia, Australia, and the Americas have a higher prevalence [5]. Additionally, 91% of lupus patients are female. The appearance of the disease before the age of 14 only occurs in 8% of patients, while the disease appears after age 50 for 9% of patients [2]. If the lupus condition is entirely cutaneous, the difference between sex decreases, and there are three females for each male [3]. The prevalence of lifetime and current self-reported comorbidity and the quality of life associated with LE leads it to be considered a chronic disease [6]. The chronic illness lasts longer than three months or requires continuous hospitalization for more than a month with an unpredictable, changing course and is so severe that it interferes with the patient’s daily activities [7]. The impact of the diagnosis, prognosis, and treatment of chronic disease, as well as the dynamics of daily family functioning due to the demands created by the disease and the appearance of possible complications, have a significant influence on the well-being and quality of life of both the patients and their families [8]. In this context, we cannot ignore that in the case of LE, insufficient control of disease activity results in trigger outbreaks of lupus, which are related to an increase in costs, with hospitalizations being the major component [9].

The chronic illness affects the quality of life [10], generates changes in both physical and emotional dimensions: loss of independence, loss of autonomy, hopelessness, and uncertainty. In this sense, quality of life is understood as a construct that unites living conditions and satisfaction with them [11]. Similarly, quality of life does not always correlate with the disease’s actual activity or damage as measured by scales of organic damage [12,13]. Quality of life is worse in patients with SLE in all the domains evaluated, except when compared with patients with depression, who present lower levels in mental health and emotional role domains than patients with SLE [14].

All these reasons lead to the fact that quality of life, in general, has recently become more relevant [8,11]. In the case of LE, this situation is all the more important because pain, physical appearance, neuropsychological impairment, the long time it takes to make an accurate diagnosis, impaired body image and self-concept lessen the quality of life of patients with LE [8,15]. Nursing care influences patient satisfaction and quality of life [16,17]. In this regard, nurses’ role as educators and health promoters should be highlighted [18]. Some nurses such as Pender, Murdaugh, and Parsons [19] propose health promotion as a tool directly related to quality of life. For this reason, it is necessary to know the quality of life of patients to be able to act from a professional point of view, both clinically and educationally, covering aspects of promotion and prevention [20,21].

After reviewing the literature, in this context, we can confirm the existence of scales to measure organic damage or disease activity: the Systemic Lupus Erythematosus Disease Activity Index (SLEDAY) [22] and Cutaneous Lupus Activity and Severity Score/Index (CLASS/ CLASSI) [23]. Others tools to assess patients’ quality of life are MOS 36-Item Short-Form Health Survey (SF-36) [24], Dermatology Life Quality Index (DLQI) [25], SLE Symptom Checklist (SSC) [26], Simple Measure of the Impact of Lupus Erythematosus in Youngsters (SMILEY) [27], and Systemic Lupus Erythematosus-Specific Quality-of-Life Instrument (SLEQOL) [28].

Specific instruments are the Systemic Lupus Erythematosus Quality of Life Questionnaire (L-QoL) [29], Lupus Quality of Life (Lupus QoL) [13], Lupus Patient-Reported Outcome tool (Lupus Pro) [14] and Skindex to measure dermatology-specific quality of life [30]. In summary, most of the scales presented in their original version are written in English and validated in Spanish. These instruments are too extensive, and do not follow the indications of experts such as García-Riaño [31], considering the multidimensional aspect of the quality of life and its effect on the patient’s biopsychosocial factors. A major limitation they present is that they measure health-related quality of life, skin pathology, and SLE [32,33]. However, we did not find instruments that evaluate LE in general, SLE and CLE, in Spanish patients. It is also recommended to perform this measurement not only taking into account scales that measure organic damage or disease activity, since these factors influence but are not determinant. This includes patients’ perspectives to facilitate a holistic approach regarding their disease and its management [34,35]. It is also important that it is developed in the society in which it will be used since the socioeconomic and historical, ethical, and political realities influence the subjects’ perception and establish the parameters assumed to be social and individual with regard to the quality of life [35,36].

In addition, the multidimensional aspect of the quality of life and the effect on the entire biopsychosocial sphere of the patients with lupus erythematosus makes it difficult to measure. These factors could be studied with a new specific questionnaire for this group of patients, in which the differences between the different types of lupus and their relationship with the quality of life could be seen, assessed with the same criteria, a questionnaire that assesses the quality of life in the broad lupus spectrum. In this way, we also minimize possible biases related to the use of different measuring instruments allowing the comparison between population groups with similar characteristics or needs [37].

In light of this, the study aimed to develop the LEQoL scale for measuring the LE patients’ quality of life in the Spanish context, including their perspectives, and to test the psychometric properties. Therefore, the aim was to assess the quality of life levels of patients with lupus erythematosus in general (SLE and CLE) as well as provide percentiles to interpret the results of the LEQoL scale. These results can help health managers identify areas of improvement in the work environment to create and maintain a strong workforce that delivers high-quality care.

## 2. Materials and Methods

A cross-sectional study was conducted with a convenience sample that included 164 patients from the AVALUS (Valencian Association of Lupus Patients), Spain. From these, 158 patients met the inclusion criteria for the study: diagnosed as systemic or cutaneous/discoid lupus erythematosus, “going to the doctor’s office,” “attending the activities of an association of lupus patients,” were included. All 158 patients were over 18 years old and signed an informed consent form. The data collection process occurred between January 2015 and January 2017.

This study respected the fundamental principles of the Declaration of Helsinki (World Medical Association 2013), with particular emphasis on the anonymization of the data collected, confidentiality, and non-discrimination of participants. The present work was authorized by the Ethics Committee of the General University Hospital of Valencia and the AVALUS. All participants received detailed information about the aim and procedures and were informed of confidentiality. Informed consent was obtained from all individual participants included in the study.

### 2.1. Design and Data Collection

To develop and psychometrically test the LEQoL scale in a convenience sample of lupus patients, the development of the instrument was carried out in four phases. (Phase 1) definition of the LEQoL factors and items and (Phase 2) development of the LEQoL. (Phase 3) content validity, and (Phase 4) psychometric evaluation of the LEQoL (Figure 1).

#### 2.1.1. (Phase 1) Definition of Quality of Life of Patients with Lupus Erythematosus Instrument (LEQoL) Factors and Items

The definition of LEQoL instrument factors and items was based on the literature review. Although no instrument was observed for Spanish patients for any type of lupus, the literature review established the following factors of the questionnaire according to the nine instruments reviewed and following the recommendations of experts such as García-Riaño and Ibáñez [31]: general health perception (4 items), physical health (19 items), pain (3 items), anxiety-depression (9 items), cognitive function (4 items), physical appearance (11 items), social activities (8 items), affective relationships (6 items), sexual relations (4 items), treatment-related effects (4 items) and economic difficulties (2 items). In each of these dimensions, a total of 74 items were formulated.

#### 2.1.2. (Phase 2) Development of LEQoL

The questionnaire (version 1) consisted of two parts, five socio-demographic questions (age, sex, occupation, time with lupus, and time to diagnosis), and 74 items scored on a Likert scale, ranging from 0 “least affected” to 10 “most affected.”

#### 2.1.3. (Phase 3) Content Validity

The instrument’s content validity was assessed in a pilot sample of 40 patients from the AVALUS and 10 experts in the field of study. The experts were nurses and doctors selected based on their clinical experience [38]. We considered this to be at least 10 years of professional experience with lupus erythematosus patients. The experts were asked to assess the accuracy, clarity, legibility, and relevance of each item of the questionnaire. The content validity index (CVI) [39] was calculated to assess the experts’ agreement on each question.

According to the patients and experts’ results, it became necessary to add 2 socio-demographic questions and 5 questions to the initial scale. In this way, version 2 of the instrument consisted of 7 socio-demographic questions and 79 items.

#### 2.1.4. (Phase 4) Psychometric Evaluation of LEQoL

The psychometric evaluation of the final instrument was performed on a new sample of 158 patients. The researcher carried out semi-structured interviews, and each interview lasted approximately 40 min.

### 2.2. Statistical Analysis

The statistical analysis was conducted in the Statistical Package for the Social Sciences, SPSS version 21.0.0 (Armonk, NY, USA) [40], and EQS (Structural Equation Modelling Software, Version 6.2) (EQS, Munich, Germany). Psychometric properties were then tested using exploratory factorial analyses (EFA), confirmatory factorial analyses (CFA), average variance extracted (AVE) [41], Cronbach’s alpha, and composite reliability (CR). First, EFA and CFA were used to validate the factorial structure of the scale using estimates provided by the robust Satorra Bentler correction estimate of maximal verisimilitude (MVR) to correct for the absence of multivariate normality. The absence of normality was analysed using Mardia’s coefficient. The adequacy of the CFAs was tested using significance calculated from the chi-square test with the robust Satorra–Bentler correction (S-B χ^2^) [42,43].

The adequacy of the proposed models was tested with additional coefficients, such as the χ^2^ ratio and its degrees of freedom (χ^2^/df) as well as the S-B χ^2^ and its degrees of freedom, with values of less than five considered acceptable [44]. The coefficients of the goodness of fit of the proposed models were tested with the non-normed fit index (NNFI), the comparative fit index (CFI), and the incremental fit index (IFI). For these indicators, values >0.90 were considered a good fit [45]. To conclude, the root mean-square error of approximation (RMSEA) was sampled. These ratings were required to be <0.08 to be considered a good fit [46]. Convergent validity was analysed using the CFA results, whereas the AVE test was used for discriminant validity [41]. After performing EFA and CFA, the items’ properties were analysed using item-total correlation coefficients and variations in the Cronbach’s alpha coefficients if items were eliminated.

## 3. Results

### 3.1. Sample Characteristics

The sample’s fundamental characteristics are 142 women (90%) and 16 men (10%), establishing a ratio of 1 man to 8.8 women. The participants’ ages ranged from 18 to 72 years (M = 42; SD = 9.61). Moreover, 73% of respondents (115) had systemic lupus erythematosus, and 27% of patients (43) had cutaneous lupus erythematosus; see Table 1 for more information.

### 3.2. Psychometric Evaluation of the Scale in the Final Sample of Patients

#### 3.2.1. Validity Analysis

The psychometric evaluation of the scale was evaluated by EFA and CFA. Previously, Bartlett’s sphericity test and Kaiser Meyer Olkin’s (KMO) index were used to verify the sample’s adequacy; low values of this index (<0.05) discourage the performance of factorial analysis. The values indicated values of KMO = 0.86 (>0.5) and the sphericity test result of Bartlett (χ^2^ = 2429.25; df = 210; *p* ≤ 0.001), so we proceeded to calculate EFA and CFA.

#### 3.2.2. Exploratory Factorial Analysis (EFA)

The 79 items were evaluated in the EFA using principal component analysis [47]. The rotation to the simple structure was obtained in an orthogonal form (varimax). After several exploratory factorial analyses, the variables were reviewed, and those that scored low (<0.3) on more than one factor were eliminated. The resulting model consisted of 21 items and 5 factors that explained 75.3% of the variance: physical, appearance, emotions, cognition, and relationships. Cronbach’s alpha measured the reliability of all factors separately and was always 0.91 and 0.92 on a total scale.

#### 3.2.3. Confirmatory Factorial Analysis (CFA)

Once EFA was performed, confirmation of the instrument’s factor structure (items grouped into 5 factors) was performed using a confirmatory factor analysis 21 using the robust correction estimate of maximal verisimilitude (MVR), as it is the most robust method of estimation with non-normal data.

The definitive model composed of 21 items grouped into five factors had adequate psychometric properties: [(S-Bχ^2^ = 352.19, df = 179, *p* < 0.01); (χ^2^ = 379.747, df = 179); (χ^2^/df = 2.12) indicated a good fit as it showed a value of less than 5; (RMSEA = 0.07; CI = 0.065–0.089); NNFI = 0.91; CFI = 0.93; and IFI = 0.93)]. The RMSEA was 0.07, which agrees with the minimal acceptable fit criteria (≤0.08).

The results justify the factorial validity of the scale. In order to increase the empirical evidence, discriminant and convergent validity were calculated. Discriminant validity was evaluated using the average variance extracted test (AVE) [41], as shown in Table 2. The AVE square root must be higher than the correlation among the pairs of factors considered in order to determine the existence of discriminant validity [41]. Convergent validity appeared to be adequate, showing a significant and strong correlation between the scale items and the latent variables that they were supposed to measure, with *t*-values over 4.26 in every case and loadings for every factor of over 0.70 on average [48].

#### 3.2.4. Item Analysis

The 21 items of the LEQoL scale were analysed. The final items mean (M), standard deviations (SD), item-total correlations (r_jx_), Cronbach’s alphas without the item (α-x), and the values for asymmetry (A) and kurtosis (K) are shown in Table 2. The contribution of every item to the scale seems to be satisfactory. The elimination of any item does not improve the reliability of the scale (0.92). In order to verify a normal item score distribution, the values for asymmetry and kurtosis were observed.

#### 3.2.5. Reliability Analysis

In order to examine scale reliability, internal consistency was calculated using Cronbach’s alpha. However, as this index does not consider the influence upon the construct reliability, both the average variance extracted (AVE) and the composite reliability coefficient (CRC) [41] were calculated. Values above 0.50 are recommended for the AVE [49], and the minimum CR value considered to be adequate is 0.70 [49]. The LEQoL scale obtained a total alpha value of 0.92. The different dimensions presented alpha values of between 0.82 and 0.92 (physical refers to situations of pain and fatigue α = 0.92; appearance reflects the importance of these patients’ outward appearance α = 0.88; emotions emphasizes the emotional response α = 0.90; cognition represents the possible difficulties at cognitive level α = 0.89; and relationship highlights the possible influence of the disease on interpersonal relationships and their maintenance α = 0.81). Moreover, the five factors presented acceptable CR and AVE values (Table 2) [41]. The instrument’s final version consisted of 7 socio-demographic questions, 21 items grouped into five factors (final instrument, Table A1).

The five factors include: physical factor implies mainly fatigue and pain, and in some cases at a renal and cardiac level, as serious manifestations. The appearance factor relates to physical problems such as weight gain, skin lesions, and how much these affect the patients’ day-to-day life. The emotions factor assesses different emotional aspects, and how many of them are manifested in patients because of their illness. The cognition factor presupposes that the patient may have affected memory, attention, reflexes, among others. Finally, the relationship factor makes known some relationship problems because of the disease, both at the level of the couple, as well as at the level of close family and social relationships. The score that each patient reflected after the interview gives an approximation of their quality of life.

### 3.3. Quality of Life Level in the Sample of Patients

After psychometric evaluation of the scale (LEQoL), patients’ quality of life with LE was analysed. First, the main descriptive elements (mean and standard deviation) of the dimensions that make up the LEQoL were analysed, followed by an analysis of the differences in these levels according to the type of lupus. The Mann–Whitney U-test was used to compare averages, and Cohen’s *d* was calculated as a measure of effect size for the comparison of two independent groups. According to Cohen [50], it is considered a small effect if *d* = 0.20, moderate effect if *d* = 0.50 and a large effect if *d* = 0.80.

In the case of patients with LE, the most widely accepted factor is the physical factor, in both types of lupus (M = 6.1; SD = 2.93) and in SLE (M = 6.65; SD = 2.52). The emotion factor is the most affected in CLE (M = 5.78; SD = 3.42) and the second most affected factor in both types of lupus (M = 5.8; SD = 2.78) and SLE (M = 5.86; SD = 2.51). The following factors affected in both types of lupus would be cognition (M = 5.2; SD = 3.2), appearance (M = 4.5; SD = 2.98), and relationships (M = 2.4; SD = 2.78). In SLE, the following factors affected are cognition (M = 5.78; SD = 2.92), appearance (M = 4.71; SD = 2.79) and relationships (M = 2.95; SD = 2.87). However, in the case of CLE, the second most affected factor would be physical (M = 4.67; SD = 3.44), and the third most affected factor would be appearance (M = 4.07; SD = 3.45), followed by cognition (M = 3.65; SD = 3.72) and relationships (M = 1.20; SD = 2.06).

Also, the differences between different lupus types are statistically significant (*p* < 0.05) in terms of physical factors, cognition, and relationships. Physical (d_Cohen_ = 0.65), cognition (d_Cohen_ = 0.63) and relationship (d_Cohen_ = 0.70) factors are affected, with a lower quality of life for patients with systemic lupus compared to patients with cutaneous lupus (Table 3).

### 3.4. Percentiles of LEQoL Interpretation

Finally, to facilitate the interpretation of the data obtained from the LEQoL scale, some percentiles were calculated based on the scores of the CLE and SLE samples. To that end, the 10, 20, 30, 40, 50, 60, 70, 80, and 90 percentiles were calculated (Table 4).

## 4. Discussion

The importance of this research is to develop an instrument for measuring the quality of life in LE patients in general and not only in the case of SLE, in the Spanish context. Moreover, it aimed to psychometrically test the instrument and assess the quality of life levels in LE patients in general, including SLE and CLE. In the literature, we find instruments that measure the quality of life in patients with SLE but not in patients with SLE and CLE: SSC [26], SMILEY [27], SLEQOL [28], L-QoL [29], Lupus QoL [13] and Lupus pro [14]. These instruments are too extensive and do not follow the indications of experts such as García-Riaño [31]: to include patients with SLE and CLE perspectives to facilitate a holistic approach regarding their disease and its management [34,35]. Moreover, most of them were originally written in English, and had not been generated in the Spanish context, without considering the parameters assumed to be social and individual on the quality of life [35,36]. On the other hand, we intended to analyze the quality of life of patients with lupus and determine percentiles for interpretation of the data.

Based on the literature review, a questionnaire was constructed consisting of 74 items grouped into 11 factors. After applying this instrument to a pilot sample and the judgment of experts, the resulting instrument was grouped into 79 items with 11 factors. The psychometric properties of the instrument were then analyzed. The instrument’s final version consists of a 21-item questionnaire grouped into 5 factors that present adequate psychometric properties. For the instrument’s characteristics following the recommendations proposed by García de Yébenes et al. [51], we can note that this is a brief, clear and self-referred questionnaire, which gives it adequate viability.

According to our results, the quality of life of patients with LE in our study is at average levels, and that quality of life is significantly worse in patients with SLE than patients with CLE. In the case of physical factors, cognition, and relationships, a moderate effect size (0.60) was observed. In the literature, we find studies comparing the quality of life of patients with CLE to those with other skin pathologies and find that quality of life is worse in CLE cases [52]. Similarly, some studies evaluate the quality of life in patients with SLE, indicating that the levels are low compared to other chronic pathologies [13,53]. However, we did not find any studies that determine the quality of life in patients with LE in general, considering both patients with CLE and SLE. We assessed the quality of life of patients with LE in general, SLE, and CLE. In addition, there is little we know about the quality of life of patients with LE [53]. In the present study, pain, disease unpredictability, and chronic fatigue, as well as the effect on memory, attention, and social relations, are the aspects that have a major impact on the quality of life. These results are in line with those proposed by authors such as Ng and Chan [54] because of their symptomatology, the unpredictable sequence of disease outbreaks, the side effects of medication, SLE patients may not feel able to establish relationships. Prevention of disease outbreaks is a cornerstone of clinical care for lupus, including patients’ perspectives on their disease and its management can help design patient-centered strategies to improve quality of life [34,35].

This instrument is the first step towards having an instrument to measure the quality of life of patients with LE in the Spanish context, considering the patients’ perspectives. This is in order to be aware of the relevance of the problem, considering the patients’ emotional situation from the beginning of the disease. It aims to assess their process of acceptance, and make an appropriate and individualized follow up of the whole process, working on coping strategies, and psychological and emotional affectation in order to design an intervention plan that incorporates these variables reducing the negative impacts and improve the quality of life in patients with LE.

Nevertheless, the study had some limitations. The subjects were exclusively from the Valencian Community, and so the results should be generalized with caution. It would be interesting to increase the sample size for future studies, evaluate the test–retest reliability and extend this study to other populations in Spain. This would include the patients’ socioeconomic status to assess the differences between SLE and CLE from this perspective. The strength of the study is that the questionnaire (LEQoL) assesses the quality of life in LE patients in general, SLE and CLE, and not only in the case of SLE.

## 5. Conclusions

Our results show that the quality of life of patients with LE in our study is at average levels and that quality of life is significantly worse in patients with SLE compared to patients with CLE. We attempt to provide more comprehensive evidence by presenting an instrument, the LEQoL scale, that has adequate reliability and validity as a useful tool for measuring an LE patient’s quality of life in the Spanish context, including SLE and CLE patients. This instrument was constructed with the contributions of 158 patients affected by LE through interviews.

The instrument has several potential applications for healthcare managers and nurses who are widely concerned about the quality of care. First, the assessment from a perspective of self-reporting may lead to the determination of training needs and to the evaluation of the effectiveness of training and interventions to improve quality of life in patients with LE, including SLE and CLE patients. Finally, the existence of the instrument and the percentiles facilitates the interpretation of scores and allows for a quick comparison with other samples of patients.

## Figures and Tables

**Figure 1 ijerph-17-08642-f001:**
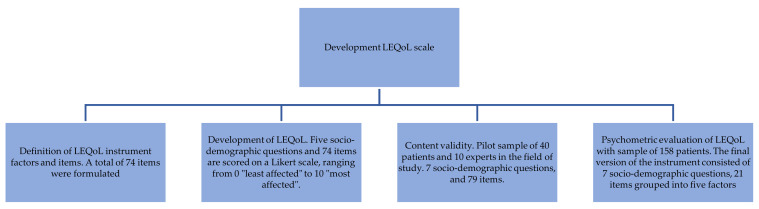
Quality of Life of Patients with Lupus Erythematosus Instrument (LEQoL) instrument development process.

**Table 1 ijerph-17-08642-t001:** Characteristics of the final sample.

Characteristics (*N* = 158)	*N*	%
**Gender**		
Women	142	90
Men	16	10
**Lupus erythematosus type**		
Systemic	115	73
Cutaneous	43	27
**Time to diagnosis**		
0 to 3 months	24	15
3 to 6 months	10	6
6 to 9 months	7	4
9 to 12 months	3	2
More than 12 months	61	39
They do not know/they do not remember	53	34

**Table 2 ijerph-17-08642-t002:** Analysis of the 21 items of the LEQoL: mean (M), standard deviation (SD), item-total correlation (r_jx_), Cronbach’s alpha if it eliminates the element (α-x), asymmetry (A), kurtosis (K), composite reliability (CR) and average variance extracted (AVE).

Complete Questionnaire (α = 0.92)	M	SD	r_jx_	α-x	A	K
**Physical α = 0.92; CR = 0.92; AVE = 0.66**						
LEQoL 11	7.12	3.29	0.67	0.91	−1.24	0.22
LEQoL 12	6.39	3.34	0.63	0.91	−0.90	0.50
LEQoL 13	7.13	3.35	0.62	0.91	−1.28	0.19
LEQoL 14	3.87	3.54	0.60	0.91	0.15	−1.50
LEQoL 24	6.51	3.51	0.59	0.91	−0.84	−0.73
LEQoL 26	5.65	3.71	0.62	0.91	−0.45	−1.30
**Emotions α = 0.91; CR = 0.89; AVE = 0.61**						
LEQoL 29a	5.75	3.18	0.58	0.91	−0.54	−0.87
LEQoL 29b	5.16	3.27	0.55	0.91	−0.26	−1.09
LEQoL 29c	6.27	3.04	0.64	0.91	−0.62	−0.57
LEQoL 29d	6.06	3.14	0.66	0.91	−0.60	−0.69
**Cognition α = 0.89; CR = 0.91; AVE = 0.71**						
LEQoL 39	5.27	3.70	0.56	0.91	−0.39	−1.37
LEQoL 40	5.37	3.71	0.60	0.91	−0.35	−1.41
LEQoL 42	4.73	3.56	0.62	0.91	−0.12	−1.43
**Appearance α = 0.88; CR = 0.90; AVE = 0.75**						
LEQoL 43	4.68	3.73	0.53	0.91	−0.02	−1.48
LEQoL 44	3.47	3.79	0.52	0.91	0.57	−1.24
LEQoL 45	3.57	3.49	0.52	0.91	0.47	−1.13
LEQoL 46	5.02	3.68	0.53	0.91	−0.14	−1.40
LEQoL 48	5.66	3.46	0.52	0.91	−0.47	−1.09
**Relationships α = 0.82; CR = 0.83; AVE = 0.63**						
LEQoL 62	2.65	3.42	0.46	0.92	0.90	−0.68
LEQoL 63	2.76	3.44	0.43	0.92	0.82	−0.82
LEQoL 66	2.03	2.96	0.35	0.92	1.30	0.42

**Table 3 ijerph-17-08642-t003:** Descriptive factor results for both types of lupus.

	Both Types of Lupus		Systemic		Cutaneous						
Factors	M	SD	M	SD	M	SD	U	*p*	*d*	MAX	MIN
Physical	6.1	2.93	6.65	2.52	4.67	3.44	1674.50	0.00 *	0.65	10	0
Appearance	4.5	2.98	4.71	2.79	4.07	3.45	2096.00	0.14		10	0
Emotion	5.8	2.78	5.86	2.51	5.78	3.42	2306.50	0.51		10	0
Cognition	5.2	3.2	5.78	2.92	3.65	3.72	1636.50	0.00 *	0.63	10	0
Relationships	2.4	2.78	2.95	2.87	1.20	2.06	1450.50	0.00 *	0.70	10	0

Note: * *p* < 0.05; U = U Mann–Whitney; d = Cohen; M = mean; SD = standard deviation; MAX = maximum; MIN = minimum.

**Table 4 ijerph-17-08642-t004:** Percentiles of LEQoL interpretation.

Factors	Systemic	Cutaneous	Total (Max. 20)	C 10	C 20	C 25	C 30	C 40	C 50	C 60	C 70	C 75	C 80	C 90
F1	6.65	4.67	11.32	0	3.63	4.3	5.3	6.5	7	7.6	8	8.1	8.3	9.3
F2	4.71	4.07	8.78	0	1.6	2	2.8	3.5	4.6	5.2	6	6.6	7.4	8.8
F3	5.82	5.78	11.6	1.5	3.5	4.25	4.6	5	6	6.7	7.7	8.2	8.7	9.25
F4	5.78	3.65	9.35	0	1.5	2.3	2.9	5	6.1	7	7.6	8	8.3	9.03
F5	2.95	1.20	4.13	0	0	0	0	0.53	1.5	2.6	3.6	4.6	5	7
Total (Max. 50)	25.91	19.37	45.18											

F1 = Physical; F2 = Appearance; F3 = Emotion; F4 = Cognition; F5 = Relationships.

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
