# Peer review of "Lupus Erythematosus Quality of Life Questionnaire (LEQoL): Development and Psychometric Properties"

_ijerph, 2020, doi:10.3390/ijerph17228642_

Round 1
Reviewer 1 Report
The study by Castellano-Rioja et al. describes an instrument to test the quality of life of patients with lupus erythematosus (LEQoL) in a cohort of the Spanish population. They further state that the by using LEQoL scale they found that the quality of life is significantly worse in patients with systemic lupus erythematosus (SLE) in comparison to cutaneous lupus erythematosus (CLE). Though this study is useful in better assessing the quality of life of lupus patients, still there are few concerns:
- The title does not fully reflect the study. I would recommend using a better title for the study.
- The study cohort includes 90% were females and only 10% males. I wonder if increasing the male percentage would change the study outcome?
- I wonder why some aspects of socio-economic factors were not taken into consideration while designing the questionnaire? Is there any difference between SLE and CLE from this perspective?
- The manuscript does not clearly describe the clinical implication of this study. In addition, the authors should comment on how they are going to improve the quality of life of lupus patients in the future using their instrument.
- Abbreviations need to be properly defined. For example, EFA, CFA, etc.
- There are some grammatical errors and complex sentences that are difficult to understand. For example, line 54-56, line103-104, line 249-251, etc. I would recommend the authors to check the language throughout the manuscript and correct those mistakes.
Author Response
Comments and Suggestions for Authors
The study by Castellano-Rioja et al. describes an instrument to test the quality of life of patients with lupus erythematosus (LEQoL) in a cohort of the Spanish population. They further state that the by using LEQoL scale they found that the quality of life is significantly worse in patients with systemic lupus erythematosus (SLE) in comparison to cutaneous lupus erythematosus (CLE). Though this study is useful in better assessing the quality of life of lupus patients, still there are few concerns:
Dear reviewer,
We would like to thank you for their time and effort in revising our paper. We really appreciate the suggestions made and we believe the document has greatly benefited from it.
In general, title, abstract, the introduction has been modified with more updated references and with the contribution of the study. The methods, results and discussion section has been changed to clarify and make it easier to read. In the method section, a figure has been added to clarify the process of instrument development. References have been reviewed, some of them have been modified to comply with their comments and the journal´s requirements. The paper has been reedited by a native English speaker. All changes have been included in the article in red.
Reviewer 1 Comments:
1. The title does not fully reflect the study. I would recommend using a better title for the study.
Response: We have modified the title to fully reflect the study. Page 1 L 2-3.
“Lupus Erythematosus Quality of Life Questionnaire (LEQoL): Development and psychometric properties".
2. The study cohort includes 90% were females and only 10% males. I wonder if increasing the male percentage would change the study outcome?
Response:
We appreciate your concern, this disease is far more prominent in females than in males, that causes the samples to be predominantly female. CLE has a clear female predominance of 3 or 4:1[1], a similar trend to the 8:1 female predominance in SLE [2]. We really appreciate the suggestions made on this regard we will consider increasing the male percentage in the sample in future research.
- Patel, J.; Robert, B.; Victoria P. W. An Update on the Pathogenesis of Cutaneous Lupus Erythematosus and Its Role in Clinical Practice. Rheumatol. Rep. 2020, 22, 1-10. https://doi.org/10.1007/s11926-020-00946-z
- Petersen, M. P.; Möller, S.; Bygum, A.; Voss, A.; Bliddal, M. Epidemiology of cutaneous lupus erythematosus and the associated risk of systemic lupus erythematosus: a nationwide cohort study in Denmark. Lupus 2018, 27, 1424-1430. https://doi.org/10.1177/0961203318777103
3. I wonder why some aspects of socio-economic factors were not taken into consideration while designing the questionnaire? Is there any difference between SLE and CLE from this perspective?
Response:
Given the complex nature of the quality of life concept, its evaluation requires multiple measures to capture the subjectivity and multidimensionality. Socioeconomic status does not appear in this analysis and can influence the quality and quantity of access to resources, among others. This can influence the quality of life of lupus patients. We appreciate this recommendation, and we hope that in future studies will include it and be more representative. Further analysis of this socioeconomic status of the participants would be justified. Page 3 L 97-100.
We included these aspects as a study limitation. Page 10 L 337-338.
4.The manuscript does not clearly describe the clinical implication of this study. In addition, the authors should comment on how they are going to improve the quality of life of lupus patients in the future using their instrument.
Response:
The introduction and discussion sections have been modified to explain the clinical implication of this study and how the results are going to improve the quality of life of lupus patients in the future using their instrument. Page 3 L 101-108 and page 10 L 317-333.
5. Abbreviations need to be properly defined. For example, EFA, CFA, etc.
Response:
EFA and CFA have been properly defined at the abstract section and all abbreviations have been revised. Page 1 L 19-20.
6. There are some grammatical errors and complex sentences that are difficult to understand. For example, line 54-56, line103-104, line 249-251, etc. I would recommend the authors to check the language throughout the manuscript and correct those mistakes.
Response:
We have checked the language throughout the manuscript. All changes have been included in the article in red.
Line 54-56 modified on page 2 L 63-66
“The chronic illness affects the quality of life [10], generates changes in both physical and emotional dimensions: loss of independence, loss of autonomy, hopelessness, and uncertainty. In this sense, quality of life is understood as a construct that unites living conditions and satisfaction with them [11]”.
Line 103-104 modified on page 3 L 130-133.
“To develop and psychometrically test the LEQoL scale in a convenience sample of lupus patients. The development of the instrument was carried out in four phases. (Phase 1) definition of the LEQoL factors and items and (Phase 2) development of the LEQoL. (Phase 3) content validity, and (Phase 4) psychometric evaluation of the LEQoL (Figure 1)”.
Line 249-251 modified on page 10 L 291-294
“The importance of this research is to develop an instrument for measuring the quality of life in LE patients in general and not only in the case of SLE, in the Spanish context. Moreover, to psychometric testing the instrument and to assess the quality of life levels in LE patients in general, including SLE and CLE”.
Reviewer 2 Report
The authores developed a new QoL questionnaire for lupus patients and provided some data on the validation proces. The process of developing of new scale is well described and the methods appriopriate. Looking at the results I would expect some more psychometric analysis, namely: test-retest comparison as it provides some information about the understanding and relevance of the scale as well as comparison of the new instrument with other QoL instruments (e.g. generic ones) or at least with disease severity scales (e.g. SLEDAI, CLASI).
Nevertheless, the study is important and a new instrument quite interesting to be further tested in clinical settings.
The authors performed their study according to standards - so the paper is methodologically sound, I do not have queries regarding the mathodology.
The number of patients included in the study is sufficient.
Maybe the authors may provide better explanation, how their instrument overcome previous shorcomings in the QoL measurement.
Author Response
Dear reviewer,
We would like to thank you for their time and effort in revising our paper. We really appreciate the suggestions made and we believe the document has greatly benefited from it.
References have been reviewed, some of them have been modified to comply with their comments and the journal´s requirements. The paper has been reedited by a native English speaker. All changes have been included in the article in red.
Reviewer 2 Comments:
The authors developed a new QoL questionnaire for lupus patients and provided some data on the validation process. The process of developing of new scale is well described and the methods appropriate. Looking at the results I would expect some more psychometric analysis, namely: test-retest comparison as it provides some information about the understanding and relevance of the scale as well as comparison of the new instrument with other QoL instruments (e.g. generic ones) or at least with disease severity scales (e.g. SLEDAI, CLASI).
“I would expect some more psychometric analysis, namely: test-retest comparison”
Response:
We really appreciate the suggestions made on this regard; we have not conducted a test-retest comparison. However, we will consider assessing test-retest reliability in future research. We included it as a study limitation. Page 10 L 336.
“I would expect it provides some information about the understanding and relevance of the scale as well as comparison of the new instrument with other QoL instruments (e.g. generic ones) or at least with disease severity scales (e.g. SLEDAI, CLASI)”
Response:
We have included some information about the understanding and relevance of the scale as well as comparison of the new instrument with other QoL instruments (e.g. generic ones) or at least with disease severity scales (e.g. SLEDAI, CLASI). Page 2-3 L 79-108.
Nevertheless, the study is important and a new instrument quite interesting to be further tested in clinical settings.
The authors performed their study according to standards - so the paper is methodologically sound, I do not have queries regarding the methodology.
The number of patients included in the study is sufficient.
Maybe the authors may provide better explanation, how their instrument overcome previous shortcomings in the QoL measurement.
Response:
We have included some information to provide better explanation, how their instrument overcome previous shortcomings in the QoL measurement. Page 2-3 L 79-108.
Reviewer 3 Report
The article is well written and interesting, overall there are no major requests.
I enjoyed reading the article, and i thought that was ready to be published; I just asked to add a paragraph on patient inclusion and exclusion criteria and the number of ethical committee. before publishing it.
Statistical analysis was well performed, limitations of the study were well assessed, so I didn, t feel the necessity to add any other comment, as I think the paper would not be easily improved.
The only question would be if this article has enough scientific importance to be granted publication priority in your journal, or if a less important journal could be selected. Nevertheless, I think this paper is ready to be published and would be difficult to improve.
After that article is in my opinion ready to be published.
Author Response
The article is well written and interesting, overall there are no major requests.
I enjoyed reading the article, and I thought that was ready to be published; I just asked to add a paragraph on patient inclusion and exclusion criteria and the number of ethical committee. before publishing it.
Statistical analysis was well performed, limitations of the study were well assessed, so I didn, t feel the necessity to add any other comment, as I think the paper would not be easily improved.
The only question would be if this article has enough scientific importance to be granted publication priority in your journal, or if a less important journal could be selected. Nevertheless, I think this paper is ready to be published and would be difficult to improve.
After that article is in my opinion ready to be published.
Dear reviewer,
We would like to thank you for their time and effort in revising our paper. We really appreciate the suggestions made and we believe the document has greatly benefited from it. All changes have been included in the article in red.
The criteria for inclusion have been included in the study. Page 3 L 117-120.
The documents of the ethics committee have been sent to the journal, they do not have a numerical code so they do not appear. Page 3 L 124-126.
Round 2
Reviewer 1 Report
I thank the authors for responding to my queries.